# Preparation, Morphology and Release of Goose Liver Oil Microcapsules

**DOI:** 10.3390/foods11091236

**Published:** 2022-04-26

**Authors:** Chunwei Li, Xiankang Fan, Yangying Sun, Changyu Zhou, Daodong Pan

**Affiliations:** 1State Key Laboratory for Managing Biotic and Chemical Threats to the Quality and Safety of Agro-Products, Ningbo University, Ningbo 315211, China; lcw15968866352@163.com (C.L.); fanxiankang2022@163.com (X.F.); 2Key Laboratory of Animal Protein Food Processing Technology of Zhejiang Province, College of Food & Pharmaceutical Sciences, Ningbo University, Ningbo 315211, China; sunyangying@nbu.edu.cn

**Keywords:** goose liver oil, microcapsules, Fourier transform infrared spectroscopy, X-ray, scanning electron microscopy

## Abstract

Goose liver oil (GLO) microcapsules were prepared by konjac glucomannan (KGM) and soybean protein isolate (SPI) for the first time as wall materials. The GLO could be effectively encapsulated, with an encapsulation efficiency of 83.37%, when the ratio of KGM to SPI was 2.9:1, the concentration of the KGM-SPI composite gel layer was 6.28% and the ratio of the GLO to KGM-SPI composite gel layer was 1:6. Fourier transform infrared spectroscopy and X-ray diffraction methods showed electrostatic interactions between KGM and SPI molecules and the formation of hydrogen bonds between the GLO and KGM-SPI wall components. The results of scanning electron microscopy showed a smooth spherical surface morphology of the microcapsules with a dense surface and no cracks. The confocal laser scanning microscopy showed that the microcapsules were homogeneous inside and no coalescence occurred. The encapsulated GLO has a significantly higher thermal and oxidative stability compared to free GLO. In the in vitro digestion experiment, 85.2% of the microcapsules could travel through gastric juice, and 75.2% could be released in the intestinal region. These results suggested that microcapsules prepared by KGM-SPI might be used as a carrier for the controlled release of GLO and could microencapsulate various oil-soluble nutrients in food products.

## 1. Introduction

Goose liver oil (GLO) is extracted from goose liver and contains a considerable amount of essential fatty acids, such as linoleic acid [1,2]. Wang et al. proved that goose fat liver can regulate the lipid metabolism, reduce blood fat, fight atherosclerosis and repair the liver tissue of hyperlipidemia rats [3]. Li et al. found that GLO can repair alcoholic liver damage [4]. Therefore, GLO is a kind of potentially functional oil that is beneficial to human body, and it has important social significance to meet the needs of different consumer groups. However, during the process and storage of the food product, polyunsaturated fatty acids may degrade and lose their flavor and nutritional values as highly unsaturated molecules [5]. In addition, the direct intake of foods containing polyunsaturated fatty acids proves to be difficult in meeting the body’s needs due to their low solubility and high oxidation rate in most food systems. Therefore, the use of microencapsulated goose liver oil was expected to solve this problem.

Microencapsulation is widely used to protect susceptive food and pharmaceutical ingredients [6]. This technology plays an important part in the food and pharmaceutical industries to mask the flavor and color of foods and pharmaceuticals, to protect the functional ingredients in foods and pharmaceuticals and to prevent oxidation [7]. For example, in order to deliver nutrient-rich fish oil to the intestine, fish oil microcapsules were prepared by using two natural polysaccharides, alginate and chitosan, as wall materials by Hui Liu et al. [6]. Pea protein and pectin were used as wall materials by Leïla Aberkane et al. to microencapsulate PUFA-rich oils to improve their oxidative stability [8]. However, GLO as a core material is far different from other materials, and studies on the microencapsulation of goose liver oil have not been reported.

Konjac glucomannan (KGM) is a linear polysaccharide that consists of β-1,4-linked glucose and mannose in a molar ratio of 1.6:1 with a small number of acetyl groups. KGM has been applied in various fields because of its excellent biocompatibility, film-forming ability, gelation behavior and biodegradability [9]. Therefore, applications of KGM in the food industry are often reported. For example, Gao et al. found that increasing the concentration or temperature of KGM shortens the gelling time and improves the elasticity of KGM gels, which can be used as a gelling agent [10]. The effect of KGM on the physicochemical properties of Aspitrigla cuculus frozen surimi at −18 °C for 50 days was investigated by Jianhua Liu et al., and it was found that KGM has a strong cryoprotective effect, and so could be used as a cryoprotectant [11]. Based on these excellent properties of KGM, it was expected to be used as a wall material for microcapsules. However, when KGM was used as a microencapsulated wall material on its own, it had a low oil retention and poor emulsification capacity, which did not meet the experimental expectations. Therefore, KGM was used in combination with other wall materials. Soy protein isolate (SPI) is a mixture of various proteins that can be obtained by de-oiling soybeans at a lower temperature. SPI contains all of the essential amino acids and can provide an excellent balance of amino acid composition. Rats consuming SPI have been reported to have lower cholesterol levels and a reduced risk of cardiovascular disease. In addition, SPI has gelling, emulsification, water retention and oil retention capabilities [12]. Its excellent oil retention and gelling properties are the key reasons for why it has been chosen as a wall component for encapsulating GLO. Hou et al. used a maltodextrin and soybean protein isolate to encapsulate GLO successfully and proved the thermal stability of GLO according to the variation in weight in the 105 °C oven [13]. However, the encapsulate efficiency was not as high as expected and the thermal stability was merely determined under the single temperature. The success of encapsulating the GLO was merely proven by the scanning electron microscopy. The research of physicochemical properties and the resistance of GLO microcapsules in a gastric acid environment were also neglected. Meanwhile, studies have not been reported on the use of KGM-SPI as a wall material for encapsulated GLO.

The purpose of this study was to prepare a kind of novel goose liver oil microcapsule (GLOM) using KGM and SPI as wall materials by freeze drying in order to improve the release of goose liver oil in the intestinal tract. The encapsulation efficiency, moisture, solubility, wettability and flowability properties of GLO microcapsules were investigated under the optimal condition. Fourier transform infrared spectroscopy and X-ray diffraction methods were used to prove the successful encapsulation. Thermogravimetry and differential scanning calorimetry methods were used to analyze the thermal stability in a larger range of temperature. The morphology of the emulsion and GLOM were observed by confocal laser scanning microscopy and scanning electron microscopy, respectively. Furthermore, the oxidative stability and the in vitro release of GLOM were also investigated.

## 2. Materials and Methods

### 2.1. Material

Goose liver was purchased from Goose Co. Ltd., Ningbo, China. KGM, SPI, sodium hydroxide (NaOH), Fluorescein isothiocyanate (FITC), Nile red, dimethyl sulfoxide (DMSO) alkaline protease, ether petroleum ether, chloroform, glacial acetic acid, potassium iodide (KI), potassium bromide (KBr), soluble starch, sodium thiosulfate (Na_2_S_2_O_3_), hydrochloric acid (HCl), trichloroacetic acid (C_2_HCl_3_O_2_), thiobarbituric acid (C_4_H_4_N_2_O_2_S), pepsase, trypsin and n-hexane were reagent grade and purchased from MACKLIN Co. Ltd., Shanghai, China.

### 2.2. Preparation Microcapsules

#### 2.2.1. Extraction of Goose Liver Oil

The goose liver sample (200 g) was cut into pieces and mixed with deionized water at a ratio of 1:1 (*w*/*v*). The mixture was homogenized and the pH value was adjusted to 9.0 by using 0.1 M NaOH solution. Then, 1% (*w*/*w*) alkaline protease was added and extraction was carried out for 20 min. Then, the mixture was centrifuged at 10,000 g at 25 °C and the supernatant was defined as GLO. The extraction yield was 42% and GLO was stored in centrifuge tubes at 4 °C for further analysis.

#### 2.2.2. Preparation of Emulsion

##### Preparation of KMG/SPI Solution

KGM/SPI solution was prepared from KGM (7.0 g) and SPI (2.4 g) in distilled water and constantly stirred for 10 min at 50 °C, which was obtained according to trial planning. Then, the emulsion was homogenized by a high-speed dispersator (XHF-D, Xinzhi Corp, Ningbo, China) at 10,000 rpm for 5 min. 

##### Preparation of SPI/GLO Emulsion

SPI (2.4g) was dissolved in distilled water for 10 min at 50 °C. Then, GLO (1.6 g) was added to the solution and the emulsion was homogenized by a high-speed dispersator (XHF-D, Xinzhi Corp, Ningbo, China) at 10,000 rpm for 5 min. 

##### Preparation of KMG/SPI/GLO Emulsion

Wall solution was prepared from KGM and SPI in distilled water and constantly stirred for 10 min at 50 °C. Afterwards, GLO was added to the solution and stirred by an electronic blender (JJ-1, Xinrui Corp, Jiangsu, China) at 50 °C. Next, the emulsion was homogenized by a high-speed dispersator (XHF-D, Xinzhi Corp, Ningbo, China) at 10,000 rpm for 5 min. The emulsions were frozen at −18 °C immediately after the homogenization procedure to prevent any coalescence or flocculation [14].

#### 2.2.3. Freeze Drying

The KGM/SPI solution and KGM/SPI/GLO emulsion were precooled at −80 °C and then put into the freeze-dryer to dry for 36 h to obtain dried KGM/SPI wall and GLOM. The prepared samples were ground into powder and stored in a plastic container at 4 °C for further experiments.

### 2.3. Experimental Design for Response Surface Methodology

According to Huietal [6], the ratio of KGM to SPI (*w*/*w*), the wall material concentration (compared to distilled water) and the ratio of KGM/SPI wall to GLO (*w*/*w*) were chosen as three influential factors (marked as A, B and C). All experiments were performed in triplicate, and the model equation was expressed as:Y = a_0_ + a_1_A + a_2_B + a_3_C + a_11_AB + a_22_AC + a_33_BC + a_12_A^2^ + a_13_B^2^ + a_23_C^2^(1)
where Y represents the predicted response variable, a_0_ represents the intercept, a_1_, a_2_, a_3_ represent the linear coefficients, a_11_, a_22_, a_33_ represent squared coefficients, a_12_, a_13_, a_23_ represent interaction coefficients and A, B, C represent the above independent variables.

### 2.4. Characterization of the Microcapsules

#### 2.4.1. Analysis by Fourier Transformed Infrared Spectrometer

Fourier transformed infrared spectrometer spectra of samples were determined by a FT/IR-4700 spectrometer (JASCO Corporation, Hachioji, Japan) at room temperature according to the method of Zhou et al. [15]. GLO, KGM, SPI, dried KGM/SPI wall and GLOM were mixed with KBr, respectively. After grinding and pressing the mixture, scanning was performed from 400 and 4000 cm^−1^ at room temperature.

#### 2.4.2. X-ray Diffraction Analysis

KGM, SPI, dried KGM/SPI wall and GLOM were tested by X-ray diffractometer (D8 ADVANCE DAVINCI) with a scan speed at 0.04°/min, and the patterns were recorded in the range of 5–50° with 2θ step size [14].

#### 2.4.3. Confocal Laser Scanning Microscopy

The morphology of the emulsion was observed by using a confocal laser scanning microscopy of LSM880 as described by Yan et al. [16] with a few revisions. Fluorescein isothiocyanate (FITC, 0.1%; *w*/*v* in DMSO) was used to stain protein and Nile red (0.1%; *w*/*v* in pure water) was used to dye the oil phase. A total of 20 µL of dye mixture was added to 1 mL of emulsion.

#### 2.4.4. Morphology of Microcapsules

The morphology of wall materials and microcapsules was analyzed by scanning electron microscopy of Hitachi SU8100 and the microcapsules were gold-plated before testing [15].

#### 2.4.5. Encapsulation Efficiency

##### Surface Oil Content

The surface oil content was measured as described by Zhou et al. [15] with a few modifications. Each 2 g powder was mixed with 30 mL petroleum ether at room temperature by violent shaking for 2 min. The solvent was filtered by filter paper and then the residues of filter paper were washed twice with 10 mL petroleum ether. Afterwards, the filter liquor was collected by a pre-weighed evaporation dish. The filtrate was evaporated to a constant weight at 60 °C in a water bath and the surface oil content was measured on the basis of the difference in the pre-weighed evaporating dish.

##### Total Oil Content

The total oil content was measured by Soxhlet extraction method [17]. Two grams of powder was added into 80 mL petroleum ether. The GLO was extracted for 6 h using a Soxhlet system. The subsequent operation was consistent with the surface oil measurement.
(2)$$\mathrm{Encapsulation}\;\mathrm{efficiency}\;(\mathrm{EE})\;(\%)=(\frac{1-\mathrm{surface}\;\mathrm{oil}}{\mathrm{total}\;\mathrm{oil}})\times 100.$$

#### 2.4.6. Moisture Content

The moisture content of GLOM was determined as described by Fernandes et al. [18]. One gram of microcapsule powder was accurately weighed and put into a dryer at 105 °C to dry for 1 h until constant weight. Moisture content (MC) was determined by the equation
(3)$$\mathrm{MC}\;(\%)=(\frac{M_{0}-M_{1}}{M_{0}})\times 100.$$
where M_0_ is the weight before drying and M_1_ is the weight after drying.

#### 2.4.7. Wettability

The wettability of microcapsule was measured as described by Jinapong et al. [19]. The 100 mL distilled water was added to a 250 mL beaker and bathed at 25 °C. Then, 0.1 g microcapsule powder was poured into the funnel placed on the burette bracket. The time required for microcapsule immersion was recorded in seconds.

#### 2.4.8. Solubility

The solubility of microcapsules was calculated gravimetrically [20]. Each 1 g microcapsule powder was mixed with 100 mL pure water and stirred by a magnetic stirrer for 5 min at room temperature. Then, centrifugation was performed at 3000 rpm for 5 min. Then, the supernatant was poured into a pre-weighed evaporating dish and dried to constant weight in a dryer at 105 °C.

#### 2.4.9. Powder Flowability Experiments

According to Dima et al. [17], the flowability properties of powder was determined as follows: 3 g (M_0_) microcapsule powder was poured into the 25 mL measuring cylinder. The volume (V_0_) was directly determined and the bulk density was measured by the equation
(4)$$\rho B\;(g/\mathrm{mL})=\frac{M_{0}}{V_{0}}\times 100.$$

Then, the microcapsule powder was tapped until the volume was constant and the final volume (Vn) was measured. The tap density was determined by the equation
(5)$$\rho T\;(g/\mathrm{mL})=\frac{M_{0}}{V_{n}}\times 100.$$

The compressibility index (CI) and the Hausner ratio (HR) were used to characterize the flowability properties of microcapsules. The CI and HR values were determined using the following equations, respectively [21,22]:(6)$$\mathrm{CI}\;(\%)=\frac{\rho T-\rho B}{\rho B}\times 100.$$
(7)$$\mathrm{HR}\;(\%)=\frac{\rho T}{\rho B}\times 100.$$

#### 2.4.10. Thermogravimetric Analysis

The thermogravimetric curves of samples were performed by using a NETZSCH TG 209 F1 Libra. According to Zhou et al. [15], each 3.0–6.0 mg GLO, the mixture of GLO and dried KGM/SPI wall and GLOM were placed in crucibles and heated at a rate of 10 °C/min from 30 to 500 °C under N_2_ flow after tablet pressing.

#### 2.4.11. Differential Scanning Calorimetry Analysis

The DSC curves of samples were studied by using a NETZSCH DSC 214 Polyma DSC21400A-8096-L. Each 3.0–6.0 mg GLO, the mixture of GLO and dried KGM/SPI wall and GLOM were placed in crucibles and scanned at a heating rate of 10 °C/min from 20 to 250 °C after tablet pressing [23].

#### 2.4.12. Oxidative Stability

GLOM and GLO were placed in an oven at 60 °C, and an appropriate amount of microcapsules and GLO was taken out of the oven every day to determine the peroxide value and thiobarbituric acid active substance.

##### Peroxide Value Determination

The peroxide value of samples was measured as described by Aghbashlo et al. and Yang et al. [5,24]. The POV values of the microcapsules and GLO were calculated by the equation
(8)$$\mathrm{POV}\;(\mathrm{meq}/\mathrm{kg}\;\mathrm{oil})=\frac{\left(S-B\right)\times 0.1269\times C}{W}\times 1000.$$

In the formula, S is the titration of samples (mL), B is the titration of blank (mL), C is the concentration of sodium thiosulfate standard titration solution (mol/L) and W is the weight of samples (g).

##### Determination of Thiobarbituric acid Reaction Substance

The thiobarbituric acid active substance (TBARS) of GLO, microcapsules and KGM/SPI wall was determined by the following method with a slight modification [25]. An appropriate amount of samples were added to TBARS reagent with the ratio of 1:5 and incubated for 1 h at 90 °C. After the mixture was cooled in ice water bath for 1 h and centrifuged (6000 rpm for 10 min), the absorbance of the upper clear phase at 532 nm was measured. The absorbance of the GLO in microcapsules was calculated by the difference in microcapsule absorbance and the wall material absorbance. A standard curve was constructed by tetraethoxypropane (R^2^ = 0.9989).

#### 2.4.13. In Vitro Release Study

The in vitro release of GLO was determined as described by Paşcalău et al. with a few revisions [26].

Each 100 mg of GLOM was added in 50 mL centrifuge tubes, together with 5 mL of simulated gastric juice and simulated intestinal juice at 37 °C, respectively. Each of two samples were extracted with 20 mL n-hexane by shaking it for 10 s at different periods of time. Then, the supernatant was kept in the dark for GLO quantification spectrophotometrically at λ = 220 nm after centrifugation (4000 rpm, 2 min), against the calibration curve previously established. The blank sample was pure n-hexane. The percentage of GLO release was calculated with the equation
(9)$$\mathrm{Cumulative}\;\mathrm{release}\;(\%)=\frac{\mathrm{Released}\;\mathrm{GLO}}{\mathrm{Total}\;\mathrm{encapsulated}\;\mathrm{GLO}}\times 100.$$

The kinetic equations were used to explain the release mechanisms of GLOM in simulated gastric juice and simulated intestinal juice. Since the simulated intestinal juice curve showed a release process and a digestion process, the release process was selected to fit the kinetic equations. The type of GLO transport through the wall materials of the microcapsules was clarified with the equation
Q (%) = k·t^n^.(10)
where Q is the cumulative percentage of GLO; t is the time GLO released; k is the release constant and n is the release exponent.

In microcapsules, release mechanisms are indicated by the value of n: n ≤ 0.43 corresponds to the Fickian diffusion; for 0.43 ≤ n < 0.85, the dominant release mechanism is the diffusion and the swelling release mechanism, and n ≥ 0.85 indicates zero order release kinetics [27,28].

#### 2.4.14. Statistical Analysis

The experiments were carried out five times and the data were expressed as mean standard deviation of all determinations. Linear regression and variance analysis of response surface test were carried out by using Design-Expert (Version 8. 0. 6) (*p* < 0.05). The statistical significance of the data was determined using one-way analysis of variance (ANOVA) by SPSS 22.0 statistical software program (SPSS Inc., Chicago, IL, USA).

## 3. Results and Discussion

### 3.1. Optimization of GLO Microcapsules

The experimental design and results are exhibited in Table 1. The ratio of KGM to SPI (A), the KGM/SPI wall concentration (B) and the ratio of the KGM/SPI wall to the core material (C) were studied in the ranges of 2–4 (*w*/*w*), 4–8 (w%) and 4–6 (*w*/*w*) according to the previous preliminary investigations. The mathematical expression of the relationship of the encapsulation efficiency (Y) from variables A, B and C is given below in the equation Y = 78.34 + 1.52A + 2.40B + 5.78C − 4.55AB − 1.40AC + 2.25BC − 2.07A^2^ − 17.62B^2^ − 1.32C^2^.

The linear factor A, B and C, and interaction term BC had a positive correlation with Y. Hence, the increase in the factors resulted in an increasing tendency in the encapsulation efficiency. The interaction terms AB and AC and quadratic terms A^2^, B^2^ and C^2^ had a negative correlation with Y, which indicated a decrease in the factors, giving a decreasing trend in the encapsulation efficiency.

As is shown in Table 2, the analysis of variance showed that this model was significant (*p* < 0.01), C and B^2^ had extremely significant effects (*p* < 0.0001), B and AB had highly significant effects (*p* < 0.01) and A, BC and A^2^ had significant effects (*p* < 0.05). Therefore, this model was sufficient in clarifying the influence of the above factors on the encapsulation efficiency, and the degree of influence was: C > B > A. The R^2^ of the model was 0.9914. 

The 3D response surface of the encapsulation efficiency affected by the ratio of KGM to SPI (A), wall material content (B) and the ratio of wall material to core material (C) are shown in Figure 1. The value of the encapsulation efficiency increased with the increasing B from approximately 4.0 to 6.5, where a higher wall material concentration leads to a better encapsulation efficiency. Further increases in B led to a lower encapsulation efficiency. Moreover, a high concentration of wall materials provides enough of a continuous phase to encapsulate the oil [29] and improves the thickness and mechanical strength of the membrane, avoiding GLO leakage and increasing the encapsulation efficiency. Highly stable emulsions with small droplets lead to a high encapsulation efficiency, and the droplet size is inversely correlated with the viscosity of emulsions. For the emulsion system, the viscosity of emulsions represents the resistance of the emulsion to the flow. Coalescence will lead to an increase in the particle size of oil droplets. The emulsion breaks down eventually and the continuous phase will not encapsulate oil, so the encapsulation efficiency will decrease after freeze drying. Increasing viscosity can avoid the coalescence by reducing the movement of oil droplets. We have carefully revised the manuscript according to the suggestions [30,31]. A low concentration of KGM results in a fragile film and a high concentration of KGM leads to the heterogeneity of the emulsion. This explains why increasing the ratio of KGM to SPI (A, from approximately 2.0 to 3.5) increased the encapsulation efficiency. Furthermore, a high ratio of core material to wall material will lead to a relatively weak wall around the oil [32], leading to a low encapsulation efficiency. The optimal process conditions were verified as follows: the ratio of KGM to SPI was 2.9:1, the wall material concentration was 6.28% the ratio of core material to wall material was 1:6 and the encapsulation efficiency could reach 83.37%.

### 3.2. Fourier Transform Infrared Spectroscopy Analysis

Fourier transform infrared spectroscopy can be used to evaluate the interaction among substances, and the formation of microcapsules can be judged from the change in the shape of samples and the intensity of the infrared adsorption peak [24]. In general, KGM consists of a large amount of -OH and CH_3_CO^−^, SPI contains all kinds of amino acids with a mass of amide linkage groups and GLO consists of a great many unsaturated bonds. All of the groups can be detected by Fourier transform infrared spectroscopy [2,33]. The Fourier transform infrared spectroscopy spectra of GLO, GLOM, the dried KGM/SPI wall, KGM and SPI is shown in Figure 2A. KGM showed characteristic peaks at 3410 (-OH stretching), 2928 (-OH stretching), 1650 (amide linkage group), 1158 (-CO stretching), 1180 (-CO stretching) and 1030 cm^−1^ (-CO stretching) (Figure 2A(a)); SPI powder showed characteristic peaks at 1655 cm^−1^ (amide linkage group) (Figure 2A(d)). The absorption peaks above could all be found in the KGM and SPI wall (Figure 2A(c)) and were weakened to different degrees. GLO spectra showed sharp characteristic peaks at 3004 (=CH stretching), 2852 (-CH stretching) and 1745 cm^−1^ (-C=O stretching) (Figure 2A(b)). The characteristic peaks that appeared above also appeared in the spectra of GLOM, and only the strengths of the characteristic peak changed, whereas the position of the characteristic peak did not change, which indicated no covalent interaction or modification between the GLO and wall material. Hence, the structure and function of GLO has not changed after encapsulation. The results indicated that no covalent bonding occurs during the encapsulation process, and that KGM and SPI interacted by strong hydrogen bonding. Wang et al. have proven that KGM and SPI interacted by a strong hydrogen bond and the interaction sites were the -OH of KGM and amide linkage group of SPI via a molecular dynamics simulation [33]. Furthermore, GLO was successfully encapsulated and the GLOM framework did not change after encapsulation.

### 3.3. X-ray Diffraction Analysis

XRD can be used to characterize the crystal phase and crystallinity. The peaks of diffraction angles represent crystal phase and the crystallinity is characterized by the strength of the peak. High and narrow peaks represent a high crystallinity and stable crystal structure [34]. The diffractograms of KGM, SPI, the dried KGM/SPI wall and GLOM are shown in Figure 2B. The KGM and SPI presented the characteristic peak at 2θ = 18.4 and 2θ = 19.2, respectively. However, when they were combined as wall materials to form the KGM/SPI wall (Figure 2B(c)), the intense peak was at 14.4, indicating the changes in crystalline phases after the interaction between KGM and SPI [23]. Similarly, it could be noted that GLOM (Figure 2B(d)) displayed the sharp peak at 2θ = 17.3 and was different from the KGM/SPI wall (Figure 2B(c)), demonstrating the presence of new crystalline phases for GLOM, which is a compact structure with the KGM/SPI wall encapsulating GLO. Furthermore, the formation of new crystalline phases also proved that GLO was successfully encapsulated by KGM and SPI. Previous studies have reported that the width of the X-ray diffraction peak is correlated with the size of the crystallite, and that the increase in the peak width is usually ascribed to an imperfect crystal. The X-ray diffraction peak width of wall materials and GLOM was more narrow than KGM and SPI, suggesting that GLOM had a more compact physical structure and stable chemical properties, which was consistent with the result of the Fourier transform infrared spectroscopy analysis, which indicated that there was a strong hydrogen bond interacting between KGM and SPI.

### 3.4. Morphology of Microcapsules

#### 3.4.1. External Structure

Scanning electron microscopy could be used to observe the morphology of surface microcapsules. The morphology of the dried KGM/SPI wall exhibited a sheet form with serious dents and folds in Figure 3(B-I), which may be caused by the damage of the wall materials in the grinding process after freeze drying and the vacuum environment in the process of photographing the electron microscope [35]. After encapsulation, the sample showed less ruptures with microcapsules embedded in the wall materials, as shown in Figure 3(B-II). Analogous morphological characteristics in oil encapsulations studied by freeze drying were also reported by [35,36]. GLOM was a regular smooth spherical shape and had no crack, as shown in Figure 3(B-III), and some small pores could be observed on the surface of some microcapsules, which might be caused by the formation of ice crystal in the process of freeze drying. It could also be concluded that the GLO microcapsule was successfully prepared in the freeze-drying process.

#### 3.4.2. Internal Structure

The microscopy of the KGM/SPI solution, SPI/GLO emulsion and KGM/SPI/GLO emulsion is shown in Figure 3A. Green fluorescence is the KGM/SPI solution and the red fluorescence represents the GLO. The KGM/SPI solution showed an irregular flocculent structure, as shown in Figure 3(A-I). As shown in Figure 3(A-III), a small-sized and a little fuzzy GLO could be observed in the blue circle, which might be caused by the thickness increment owing to the addition of KGM. In addition, the irregular flocculent structure also appeared in the red circle. As shown in Figure 3(A-II), the particle size distribution of GLO was not uniform in the SPI emulsion. While the GLO appeared, a homogeneous state with no coalescence occurred and wall materials formed a shell around it. In addition, previous studies reported that the structure of the shell-encapsulating core helped to increase the encapsulation efficiency of microcapsules. Furthermore, wall materials with a homogeneous thickness could form a physical barrier, which might contribute to enhancing its oxidative stability and increasing the mechanical strength of microcapsules [24].

### 3.5. Physicochemical Properties of GLOM

The physicochemical properties of GLOM are displayed in Table 3. The moisture of the microcapsules was 3.09 ± 0.08%, the wettability was 290 ± 8.22 s and the solubility was 46.84 ± 0.79%. Wettability represents the time required for the microcapsule to immerse in water, and it represents the ability of a microcapsule to interact with water molecules [18]. Solubility is a significant indicator of the reconstitution ability of powder [37]. The microcapsules adsorbed water because of the large amount of hydrophilic groups (-OH) of KGM, increasing the adsorption capacity of water molecules on the surface of microcapsules.

Bulk density is a momentous parameter during the processing, storage and transportation conditions of powdered food [38]. The bulk density was 0.43 ± 0.01 g/cm^3^, which was higher than the bulk density of pumpkin seed oil microcapsules (0.158 g/cm^3^) by freeze drying and similar to the bulk density of flaxseed oil microcapsules (0.31–0.35 g/cm^3^) by freeze drying [22,39], indicating that GLOM with the KGM/SPI wall has advantages during the processing, storage and transportation conditions. Samples with lower bulk density values are more sensitive to oxidative degradation during storage because there are more air spaces between particles [40]. Therefore, the components we selected might contribute to the oxidative stability of GLOM. Tapping may be related to flow problems of powders, and the tap density was 0.64 ± 0.02 g/cm^3^ [41]. The compressibility index determines the cohesiveness among powders and demonstrates the capacity of microcapsules to aggregate, and the Hausner ratio represents the friction force between powders [17]. The CI and HR were 33.34 ± 1.17% and 1.50 ± 0.02, indicating that the microcapsules were ‘very poor’ in flowability (CI = 32–37; HR = 1.46–1.59) [42]. Previous studies have reported the problem of poor flowability microcapsules by freeze drying [22,35]. The result that GLOM had a good solubility may be conducive to release and digestion in simulated gastric juice and simulated intestinal juice. In addition, the bulk density index revealed that GLOM may be less sensitive to oxidative degradation during storage.

### 3.6. Thermogravimetric Analysis

The thermogravimetric method is a thermal analysis method used to measure the relationship between the mass of a substance and temperature (or time), which can be used to evaluate the thermal stability of macromolecules. As shown in Figure 4, GLO underwent one step of weight loss (Figure 4c), whereas the mixture of GLO and the dried KGM/SPI wall (Figure 4b) and GLOM (Figure 4a) showed three steps of weight loss. No weight loss occurred below 230 °C. Almost a 100% mass loss of oil with complete decomposition was observed in the range from 350 to 450 °C, and the maximal weight loss was observed at 410 °C. For the mixture, the first step of weight loss began from 30 to 180 °C, whereas GLOM lost weight from 78 to 210 °C, and the second step of weight loss began at 180 °C and 210 °C, respectively, which was ascribed to the loss of bound water [43]. The main weight loss occurred from 180 to 400 °C, which could be attributed to the dehydration and decomposition of the protein. For the final step of weight loss, the mixture started at 410 °C and GLOM started at 430 °C. Compared to the mixture, all steps of the weight loss curve of GLOM showed hysteresis. Similar phenomena and results have been found among walnut oil microcapsules [15]. It could be concluded that the microcapsules had a strong heat resistance, and that KGM/SPI wall materials enhanced the thermal stability of GLO.

### 3.7. Differential Scanning Calorimetry Analysis

Differential scanning calorimetry can be used to detect the transformation heat of a sample in a range of temperatures, which can determine the thermal stability of the sample. The differential scanning calorimetry curves of GLO, the mixture of GLO and the dried KGM/SPI wall and GLOM are shown in Figure 4B. GLO (c) underwent two phase transitions during heating: the transition from the solid phase to liquid phase began from 25 to 49 °C and the transition from the liquid phase to gas phase occurred from 172 to 207 °C. The initial temperatures of the mixture for the two phase transitions were 32 °C and 88 °C, respectively. The GLOM (a) also underwent two phase transitions during the heating process, and the initial temperatures were 62 °C and 108 °C, respectively. The first phase transition was the fusion of GLO and the second phase transition was caused by the decomposition of SPI and KGM. The fusion temperature of GLO increased from 32 to 62 °C after encapsulation. Furthermore, compared with the mixture, the decomposition temperature of SPI and KGM in microcapsules also increased from 88 to 108 °C. The result proved that GLOM remained stable below 62 °C. Generally, the temperature of the manufacture process is below 62 °C. Therefore, the physicochemical property of GLOM will not change and the nutrition of GLOM will not degrade during the manufacture process. It is concluded that, compared with GLO, the thermal stability of GLOM improved after encapsulation by the KGM/SPI wall [33].

### 3.8. Oxidative Stability Analysis

The POV and TBARS value curves are shown in Figure 5. The POV of GLO and TBARS showed a significant upward trend, and the oxidation rate was the fastest on the fourth day during the process of accelerated oxidation at 60 °C. The POV and TBARS of GLO were approximately four times that of the initial value, whereas the POV and TBARS of the microcapsules remained obviously unchanged after accelerated oxidation for 7 days, which might be due to the great antioxidant capacity of the KGM and SPI wall [23]. The results suggested that the microcapsules had an excellent antioxidant effect, which was consistent with the result of the internal structure, and the oxidation stability of GLO was significantly enhanced after microencapsulation. The antioxidant microcapsule for algal oil was prepared by the complex coacervation of soy protein with chitosan, and the TBARS value was approximately twice its initial value [44].

### 3.9. In Vitro Release Study

The in vitro release curves of GLO from the microcapsules in simulated gastric juice and simulated intestinal juice are shown in Figure 6. The original burst release occurred for the first 60 min, and 89.0% of GLO was protected in simulated gastric juice and 61.7% of GLO was released in simulated intestinal juice, respectively, which was ascribed to the soluble wall materials on the surface of the GLOM and GLO adhering to the internal surface of the wall materials, where, the more the polymer dissolved near the surface, the more GLO released [45]. The second phase showed a slow release (to 120 min), where GLO was released up to 72.7% in simulated intestinal juice but 84.6% of GLO was retained in simulated gastric juice, because the polysaccharide network protected the protein from the acid environment and pepsin degradation in the stomach [46]. However, after 120 min to 180 min, the GLO in simulated gastric juice was released and thus reduced down to 16.2%, which was caused by the digestion of GLO released in simulated gastric juice.

The kinetic equations of the in vitro release are shown in Table 4, where the Higuchi model fitted the most for the GLO release from the microcapsules in simulated gastric juice (R^2^ = 0.85244) and simulated intestinal juice (R^2^ = 0.95439). The release exponent n = 0.30834 in simulated gastric juice explained the Fickian diffusion transport mechanism, whereas, in simulated intestinal juice, n = 0.67192 displayed a non-Fickian transport mechanism. Therefore, the rate of GLO release in simulated intestinal juice is faster than that in simulated gastric juice. The result revealed that GLOM is utilized better in simulated intestinal juice, paving the way for subsequent in vivo experiments.

## 4. Conclusions

In this study, goose liver oil microcapsules were prepared by KGM and SPI for the first time as wall materials. They could be effectively encapsulated, with an 83.37% encapsulation efficiency. Electrostatic interactions existed between KGM and SPI molecules, and the formation of hydrogen bonds occurred between the GLO and KGM-SPI wall components. This indicated the successful preparation of microcapsules of encapsulated goose liver oil. Thermal and oxidative stabilities of goose liver oil were enhanced after encapsulation. Most importantly, 85.2% of the microcapsules prepared by using these two materials can pass through the gastric juice and 75.2% can be released in the intestine, achieving the goal of protecting the goose liver oil passing through the gastric juice and releasing it in the intestine. These results suggested that microcapsules prepared with KGM-SPI could be used as carriers for the controlled release of goose liver oil and other oils.

## Figures and Tables

**Figure 1 foods-11-01236-f001:**
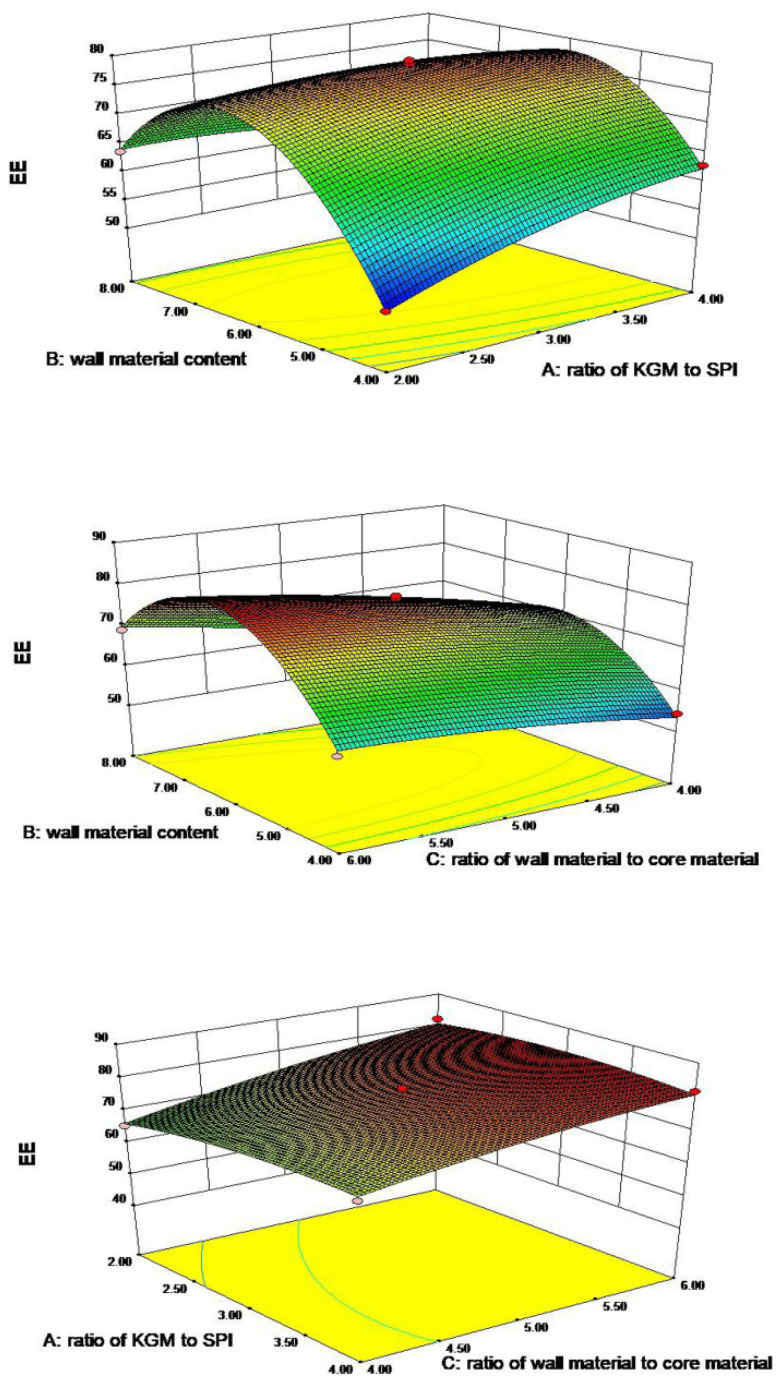
Three-dimensional response surface of encapsulation efficiency affected by the ratio of KGM to SPI (**A**), wall material content (**B**) and the ratio of wall material to core material (**C**).

**Figure 2 foods-11-01236-f002:**
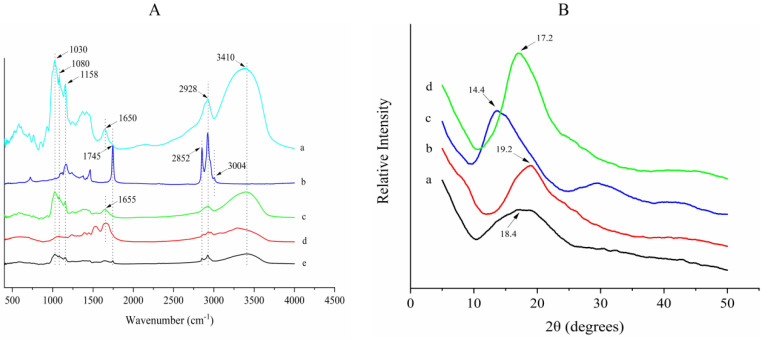
Fourier transformed infrared spectrometer spectra (**A**) of KGM (a), GLO (b), KGM/SPI wall (c) SPI (d) and GLOM (e), and X-ray diffraction patterns (**B**) of KGM (a), SPI (b), KGM/SPI wall (c) and GLOM (d).

**Figure 3 foods-11-01236-f003:**
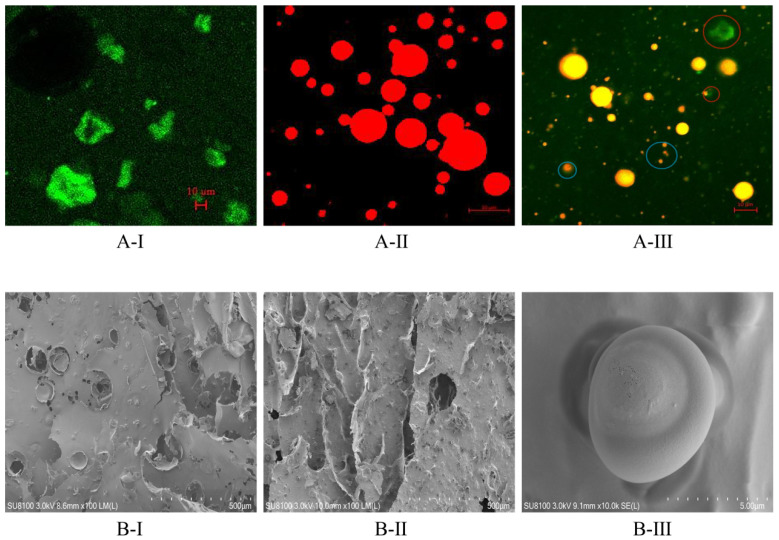
Confocal laser scanning microscopy images of KGM/SPI solution (**A-I**), SPI/GLO emulsion (**A-II**) and KGM/SPI/GLO emulsion (the intermediary emulsion prior to freeze-drying) (**A-III**), and scanning electron microscopy images of dried KGM/SPI wall (×100) (**B-I**), GLOM (×100) (**B-II**) and GLOM (×10 k) (**B-III**).

**Figure 4 foods-11-01236-f004:**
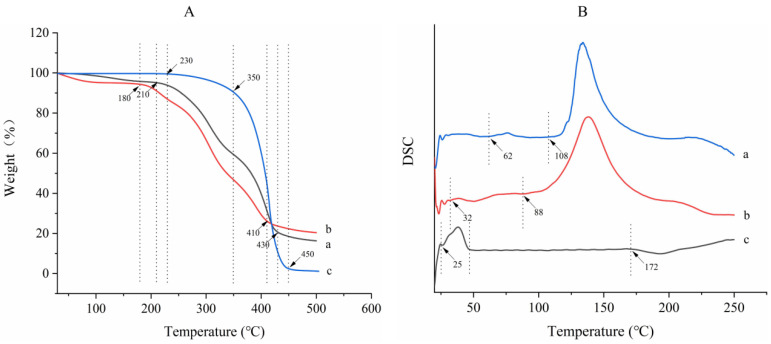
Thermogravimetric spectra (**A**) and differential scanning calorimetry spectra (**B**) of GLOM (a), the mixture of GLO and dried KGM/SPI wall (b) and GLO (c).

**Figure 5 foods-11-01236-f005:**
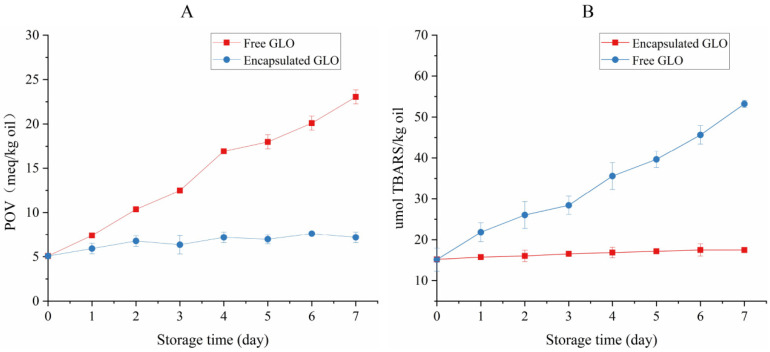
Peroxide value (POV) (**A**) and TBARS value (**B**) of encapsulated GLO and free GLO at 60 °C for 7 days.

**Figure 6 foods-11-01236-f006:**
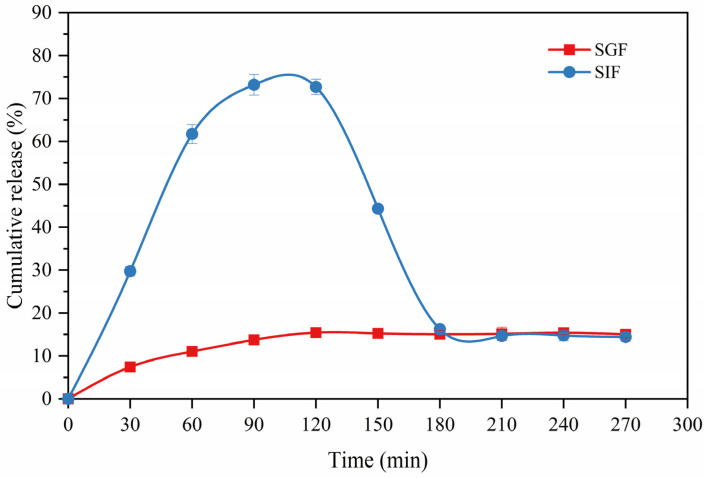
Cumulative release of GLOM in simulated gastric juice and simulated intestinal juice.

**Table 1 foods-11-01236-t001:** Box–Behnken experimental design and results.

Run	A	B (%)	C	EE (%)
1	3	4	4	54.3 ± 0.4
2	2	6	4	65.4 ± 0.6
3	4	6	4	70.8 ± 1.2
4	2	6	6	81.9 ± 0.4
5	3	8	4	55.1 ± 0.9
6	4	8	5	58.0 ± 0.5
7	3	4	6	59.2 ± 0.9
8	2	4	5	50.2 ± 0.3
9	3	6	5	78.3 ± 1.0
10	4	6	6	81.7 ± 0.7
11	3	6	5	77.0 ± 0.8
12	2	8	5	63.6 ± 0.6
13	3	8	6	69.0 ± 0.6
14	4	4	5	62.8 ± 0.7
15	3	6	5	79.2 ± 0.5
16	3	6	5	77.4 ± 0.3
17	3	6	5	79.8 ± 0.6

Note: A means the ratio of KGM to SPI, B means the wall materials concentration (%), C means the ratio of wall materials to core materials.

**Table 2 foods-11-01236-t002:** ANOVA for the response surface Box–Behnken quadratic model.

Source	Sum of	df	Mean	F Value	*p*-Value	
	Squares		Square		Prob > F	
Model	1812.22	9	201.36	89.77	<0.0001	**
A: ratio of KGM to SPI	18.61	1	18.61	8.29	0.0237	**
B: wall material content	46.08	1	46.08	20.54	0.0027	**
C: ratio of wall material to core material	266.81	1	266.81	118.94	<0.0001	**
AB	82.81	1	82.81	36.92	0.0005	**
AC	7.84	1	7.84	3.5	0.1038	ns
BC	20.25	1	20.25	9.03	0.0198	**
A^2^	18.04	1	18.04	8.04	0.0252	**
B^2^	1307.22	1	1307.22	582.76	<0.0001	**
C^2^	7.34	1	7.34	3.27	0.1135	ns
Residual	15.7	7	2.24			
Lack of fit	10.15	3	3.38	2.44	0.2046	ns
Pure error	5.55	4	1.39			
Cor total	1827.92	16				

Note: ** means significant, ns means not significant.

**Table 3 foods-11-01236-t003:** Physicochemical properties of GLO microcapsules.

Items	Index
Moisture content (%)	3.09 ± 0.08
Solubility (%)	46.84 ± 0.79
Wettability (s)	290 ± 8.22
Bulk density (g/cm^3^)	0.43 ± 0.01
Tapped density (g/cm^3^)	0.64 ± 0.02
Compressibility index (%)	33.34 ± 1.17
Hausner ratio	1.50 ± 0.02

**Table 4 foods-11-01236-t004:** Kinetic release parameters of GLO microcapsules in SGF and SIF.

Mathematical	SGF	SIF
Zero order	Q = 0.04348t + 6.48855 (R^2^ = 0.61047)	Q = 0.6294t + 9.702 (R^2^ = 0.87654)
First order	ln (100 − Q) = −4.79832 × 10^−4^t + 4.53661 (R^2^ = 0.62185)	ln (100 − Q) = −0.01186t + 4.53157 (R^2^ = 0.91876)
Higuchi	Q = 0.9196t^1/2^ + 2.63486 (R^2^ = 0.85244)	Q = 7.27531t^1/2^ − 1.51793 (R^2^ = 0.95439)
Peppas	lnQ = 1.10912 + 0.30834lnt (R^2^ = 0.81587; *n* = 0.30834)	lnQ = 1.20442 + 0.67192lnt (R^2^ = 0.89131; *n* = 0.67192)

## Data Availability

Data is contained within the article.

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
