# Peer review of "Preparation, Morphology and Release of Goose Liver Oil Microcapsules"

_foods, 2022, doi:10.3390/foods11091236_

Round 1
Reviewer 1 Report
The manuscript entitled “A kind of Novel Goose Liver Oil Microcapsule” well described the encapsulation of Goose liver oil, this was also supported by the significant results and also by in vitro digestion experiment by scanning electron microscopy. The research question in well developed and also results suggested the utility of these microcapsules for different purposes.
Therefore, there are some minor errors to check in the text:
Lines 82-86: You should add the chemical formula of each reagent used.
Furthermore, in materials and methods you miss some reagents or solvents used for the analysis. Check them and add in the text.
Line 100: you should fix “r/min” in rpm or g.
Line 167: please check the punctuation between “follows and 3g”
Line 203: you should add the coefficient of the calibration curve R2.
Please check the English language in the overall manuscript.
Reviewer 2 Report
The manuscript foods-1678321, “A kind of Novel Goose Liver Oil Microcapsule” proposes an interesting experimental study on the interest of microencapsulating Goose liver oil (GLO), also known as “goose foie gras” by using konjac glucomannan (KGM) and soybean protein isolate (SPI).
It presents the encapsulation technique, the chemical, physicochemical and morphological properties of the core-shell microcapsules (with GLO core and KGM/SPI shell). The microcapsules stability and an in vitro release of GLO in two simulated biological fluids (gastric juice and intestinal juice) is also presented.
The topic presents a clear scientific interest, considering the potential of GLO for the human health and the necessity to encapsulate it for oral administration. The characterization methods are appropriate, and the experimental measurements have been generally interpreted considering existing data in the literature.
However, I consider the manuscript needs major revision with regard to numerous points. Please find below a non-exhaustive list of elements that needs to be revised:
- The logical flow of presentation of the different characteristics should be reorganized and /or reformulate, e.g.:
Please explain on the Introduction part the (1) interest of the GLO for human health, with references. Lines 30-31 are not convincing nor scientifically accurate “Therefore, GLO is a kind of functional oil which is beneficial to human body, and it has important social significance to meet the needs of different consumer groups“
Please emphasis the originality of the present work and methodology vs. other existing GLO microcapsules, for instance the ones from the Chinese patent CN102871142A [2013], about freeze-dried GLOM, with GLO as core and a mixture of maltodextrin and soybean protein isolate as shell, and the ones from the Chinese patent CN103284158B [2014], concerning goose oil diacylglycerol microcapsules, with the GO diacylglycerol as core and a mixture of beta-cyclodextrin and sodium caseinate as shell.
The Materials and methods part presents in 2.2.2 the "Preparation of [only one] emulsion", with a given KGM /SPI/GLO composition (without explaining the choice), while in part 3.1 the "Optimization of GLO microcapsules" is presented as based on the analysis of 17 types of emulsions, with different KGM/SPI and core/shell ratios. Besides, some other parts of the manuscript discuss experimental results on samples for which there is no clear preparation methodology (e.g., KGM/SPI/GLO emulsion, SPI/GLO emulsion, KGM/SPI emulsion, KGM and SPI wall materials, etc.)
In part 2.4.6, the experimental protocol states to "put into a dryer at 105°C to dry until constant weight", without mentioning whether the three GLOM components are thermally stable at 105°C and for how long (was the drying time in the order of minutes, hours, days?).
- The results must be clearly presented, for instance:
Figures must have complete, clear and consistent captions throughout the manuscript
- Eg., the legend of Figure 3 includes "KGM/SPI emulsion", "SPI/GLO emulsion" and "KGM/SPI/GLO emulsion"; but (1) KGM/SPI mixture is not in the form of an emulsion (either solution or dried film); (2) nowhere is there any mention of "SPI/GLO emulsion"; (3) the expression "KGM/SPI/GLO emulsion" is not clear – is it the intermediary emulsion prior to freeze-drying?
- The legend of Figure 4 talks about "the mixture of GLO and wall materials" and “GLOM”. Please clearly indicate the difference in preparation of these two types of samples
Interpretation of the experimental results generally stays superficial, mostly based on citing different findings from the literature. The experimental data should be explained completely and coherently and only then to be compared with comparable literature data.
Please explain the why [lines 257-258] “Highly stable emulsions with small droplets lead to high encapsulation efficiency [29]“ AND “The droplet size is inversely correlated with the viscosity of emulsions [30].” Please also reformulate or explain the meaning of expression “viscosity of emulsions”. Besides, in the second sentence, it about the viscosity of the major phase or the ratio between the viscosities of the two media?
Please present the chemical structures of KGM, SPI and GLO before discussing the FTIR-characteristic peaks (in the part 3.2).
Please explain which kind of novel chemical groups or other changes in FTIR spectra could be expected in case of KGM, SPI and/or GLO degradation? The final conclusion from FTIR interpretation [lines 287-288] should be soundly justified (“indicated no interaction or modification between the GLO and wall material. Hence, the structure and function of GLO has not changed after encapsulation.”)
Please justify the sentence “KGM and SPI interacted by strong hydrogen bonding“ [line 290]. Which proofs or literature data?
Please explain [209-291] this conclusion “Furthermore, GLO was successfully encapsulated and the GLOM framework did not change after encapsulation“ in the light of the experimental results presented in this part of the manuscript
The X-ray diffraction analysis is rather descriptive and too general. Please indicate which kind of crystalline phases were expected vs. were obtained, and explain the “new crystalline phases for GLOM”
In the part 3.4, “Morphology of microcapsules”, before the part [lines 313-314] “the morphology of the KGM and SPI wall materials exhibited a sheet form with serious dents and folds“, please clearly indicate how these KGM + SPI samples have been prepared by freeze-drying. Normally, by freeze-drying a homogeneous KGM + SPI solution one may expecting to obtain flakes.
Please reformulate and explain the lines 333-337: “Besides, previous studies reported that core shell structure helped to increase the encapsulation efficiency of microcapsules. Besides, wall materials with even thickness properties helped to enhance its oxidative stability and to increase the mechanical strength of microcapsules [23], which indicated that GLOM might have excellent oxidative stability.”
Please explain the meaning of the comparison from lines 352-354, of the bulk density of GLOM with the ones of pumpkin seed oil microcapsules and flaxseed oil microcapsules (all obtained by freeze drying), since the bulk density of one kind of core-shell microcapsules also depends on the bulk densities of the components, the core/shell ratio, shell thickness.
Please explain which is the practical interest of the DSC study on these microcapsules vs. individual components and the interest of the different phase transitions identified by DSC
The experimental methodology and results should be systematically compared with comparable literature data.

Author Response
Q1: Please explain on the Introduction part the (1) interest of the GLO for human health, with references. Lines 30-31 are not convincing nor scientifically accurate “Therefore, GLO is a kind of functional oil which is beneficial to human body, and it has important social significance to meet the needs of different consumer groups.
Response1: Thank you for the comments. We have carefully revised the introduction according to the suggestions. See it in the line 30-33.
Wang et al proved that goose fat liver can regulate the lipid metabolism, reduce blood fat, fight atherosclerosis and repair liver tissue of hyperlipidemia rats [3]. Li et al found that GLO can repair alcoholic liver damage [4]. Therefore, GLO is a kind of potentially functional oil which is beneficial to human body, and it has important social significance to meet the needs of different consumer groups.
Q2: Please emphasis the originality of the present work and methodology vs. other existing GLO microcapsules, for instance the ones from the Chinese patent CN102871142A [2013], about freeze-dried GLOM, with GLO as core and a mixture of maltodextrin and soybean protein isolate as shell, and the ones from the Chinese patent CN103284158B [2014], concerning goose oil diacylglycerol microcapsules, with the GO diacylglycerol as core and a mixture of beta-cyclodextrin and sodium caseinate as shell.
Response2: Thanks for your advice. The introduction has been carefully revised in the line 71-78 and 84-85.
Hou et al used maltodextrin and soybean protein isolate to encapsulate GLO successfully and proved the thermal stability of GLO according to the variation of weight in the 105 ℃ oven [13]. However, the encapsulate efficiency was not as high as expected and the thermal stability was merely determined under the single temperature. The succeed of encapsulating the GLO was merely proved by the scanning electron microscopy. The research of physicochemical properties and the resistance of GLO microcapsules in gastric acid environment were also neglected. Meanwhile, the studies have not been reported on the use of KGM-SPI as a wall material for encapsulated GLO.
Thermogravimetry and differential scanning calorimetry methods were used to analyze the thermal stability in a larger range of temperature.
Q3: The Materials and methods part presents in 2.2.2 the "Preparation of [only one] emulsion", with a given KGM /SPI/GLO composition (without explaining the choice), while in part 3.1 the "Optimization of GLO microcapsules" is presented as based on the analysis of 17 types of emulsions, with different KGM/SPI and core/shell ratios. Besides, some other parts of the manuscript discuss experimental results on samples for which there is no clear preparation methodology (e.g., KGM/SPI/GLO emulsion, SPI/GLO emulsion, KGM/SPI emulsion, KGM and SPI wall materials, etc.)
Response3: Thanks very much for your suggestion, the preparation of KGM/SPI solution, SPI/GLO emulsion and KGM/SPI/GLO emulsion have been revised in our manuscript in the line 107-127.
2.2.2.1 Preparation of KMG/SPI solution
KGM/SPI solution was prepared from KGM (7.0 g) and SPI (2.4 g) in distilled water and constantly stirring for 10 min at 50℃, which has been obtained according to trial planning. Then the emulsion was homogenized by a high-speed dispersator (XHF-D, Xinzhi Corp, Ningbo, China) at 10000 rpm for 5 min.
2.2.2.2 Preparation of SPI/GLO emulsion
SPI (2.4g) was dissolved in distilled water for 10 min at 50℃. Then GLO (1.6g) was added to the solution and the emulsion was homogenized by a high-speed dispersator (XHF-D, Xinzhi Corp, Ningbo, China) at 10000 rpm for 5 min.
2.2.2.3 Preparation of KMG/SPI/GLO emulsion
Wall solution was prepared from KGM and SPI in distilled water and constantly stirring for 10 min at 50℃. Afterwards, GLO was added to the solution and stirred by an electronic blender (JJ-1, Xinrui Corp, Jiangsu, China) at 50℃. Next the emulsion was homogenized by a high-speed dispersator (XHF-D, Xinzhi Corp, Ningbo, China) at 10000 rpm for 5 min. The emulsions were frozen at -18℃ immediately after the ho-mogenization procedure to prevent any coalescence or flocculation [14].
Q4: In part 2.4.6, the experimental protocol states to "put into a dryer at 105°C to dry until constant weight", without mentioning whether the three GLOM components are thermally stable at 105°C and for how long (was the drying time in the order of minutes, hours, days?).
Response4: Thanks for your advice. Oil will degrade at the temperature over 180°C and glycerol will decompose to allyl aldehyde at the temperature over 200°C. The experiment temperature would not affect the thermal stable of the components according to the subsequent experiment. In addition, the drying time is one hour. See it in the line 174-175 “1 g microcapsule powder was accurately weighed and put into a dryer at 105℃ to dry for 1 hour until constant weight”.
Q5: The legend of Figure 3 includes "KGM/SPI emulsion", "SPI/GLO emulsion" and "KGM/SPI/GLO emulsion"; but (1) KGM/SPI mixture is not in the form of an emulsion (either solution or dried film); (2) nowhere is there any mention of "SPI/GLO emulsion"; (3) the expression "KGM/SPI/GLO emulsion" is not clear – is it the intermediary emulsion prior to freeze-drying?
Response5: Thanks very much for your suggestion, the expression of KGM/SPI solution and KGM/SPI/GLO emulsion have been revised in our manuscript. The part of SPI/GLO emulsion has been added in material and method. See it in the line 387-388 and 107-122.
Figure 3. Confocal laser scanning microscopy images of KGM/SPI solution A-Ⅰ, SPI/GLO emulsion A-Ⅱ and KGM/SPI/GLO emulsion (the intermediary emulsion prior to freeze-drying) A-Ⅲ and Scanning electron microscopy images of dried KGM/SPI wall (x 100) B-Ⅰ, GLOM (x 100) B-Ⅱ and GLOM (x 10 k) B-Ⅲ.
2.2.2.2 Preparation of SPI/GLO emulsion
SPI (2.4g) was dissolved in distilled water for 10 min at 50℃. Then GLO (1.6g) was added to the solution and the emulsion was homogenized by a high-speed dispersator (XHF-D, Xinzhi Corp, Ningbo, China) at 10000 rpm for 5 min.
Q6: The legend of Figure 4 talks about "the mixture of GLO and wall materials" and “GLOM”. Please clearly indicate the difference in preparation of these two types of samples.
Response6: Thanks for your professional suggestions. We have articulated the definition of GLO and GLOM. GLO means goose liver oil and GLOM means goose liver oil microcapsule. See the line 205-206 “The mixture of GLO and wall materials” has been revised into “the mixture of GLO and dried KGM/SPI wall”. GLOM was prepared from GLO emulsion by freeze drying. And the mixture of GLO and KGM/SPI wall were mixed by GLO and dried KGM/SPI wall with no combination.
2.2.3 Freeze drying
The KGM/SPI solution and KGM/SPI/GLO emulsion were precooled at -80℃ and then put into the freeze-dryer to dry for 36 h to obtain dried KGM/SPI wall and GLOM.
Q7: Please explain the why [lines 257-258] “Highly stable emulsions with small droplets lead to high encapsulation efficiency [29] AND “The droplet size is inversely correlated with the viscosity of emulsions [30].” Please also reformulate or explain the meaning of expression “viscosity of emulsions”. Besides, in the second sentence, it about the viscosity of the major phase or the ratio between the viscosities of the two media?
Response7: Thanks very much for your suggestion and question. Viscosity of emulsions represents the resistance of emulsion to flow. For the emulsion system, coalescence will lead to the increase of particle size of oil droplets. The emulsion breaks down eventually and continuous phase will not encapsulate oil, so that the encapsulation efficiency will decrease after freeze drying. Increasing viscosity can avoid the coalescence by reducing the movement of oil droplets. We have carefully revised the manuscript according to the suggestions. See it in line 290-295.
Q8: Please present the chemical structures of KGM, SPI and GLO before discussing the FTIR-characteristic peaks (in the part 3.2).
Response8: Thanks very much for your suggestion. The chemical structures of KGM, SPI and GLO have been presented before discussing the FTIR-characteristic peaks. See the line 315-318.
In general, KGM consists of a large of -OH and CH3CO-, SPI contains all kinds of amino acid with a mass of amide linkage group and GLO consists of a great many unsaturated bond. All of the groups can be detected by Fourier transform infrared spectroscopy [2,33].
Q9: Please explain which kind of novel chemical groups or other changes in FTIR spectra could be expected in case of KGM, SPI and/or GLO degradation? The final conclusion from FTIR interpretation [lines 287-288] should be soundly justified (“indicated no interaction or modification between the GLO and wall material. Hence, the structure and function of GLO has not changed after encapsulation.”
Response9: Thanks very much for your suggestion and question. We did not study the group changes in case of degradation. Instead, we studied the combination among KGM, SPI and GLO by the changes of characteristic peak. And we have revised our manuscript. See the line 326-328 “The characteristic peaks appeared above also appeared in the spectra of GLOM and only the strengths of characteristic peak changed but the position of characteristic peak did not change, which indicated no covalent interaction or modification between the GLO and wall material. Hence, the structure and function of GLO has not changed after encapsulation”.
Q10: Please justify the sentence “KGM and SPI interacted by strong hydrogen bonding [line 290]. Which proofs or literature data?
Response10: Thanks for your professional question. See it in the line 332-334.
Wang et al has proved that KGM and SPI interacted by strong hydrogen bond and the interaction sites were the -OH of KGM and amide linkage group of SPI via molecular dynamics simulation. The reference has been added in our manuscript.
Q11: Please explain [209-291] this conclusion “Furthermore, GLO was successfully encapsulated and the GLOM framework did not change after encapsulation in the light of the experimental results presented in this part of the manuscript.
Response11: Thanks very much for your professional question. FTIR result showed that, only the strengths of characteristic peak changed but the position of characteristic peak did not change, indicating none covalent interaction among three components. Therefore, three components of GLOM could remain primary property and structure. We have revised our manuscript and see it in the line 326-329.
Q12: The X-ray diffraction analysis is rather descriptive and too general. Please indicate which kind of crystalline phases were expected vs. were obtained, and explain the “new crystalline phases for GLOM.
Response12: Thanks very much for your professional advice. XRD can be used to characterize the crystal phase and crystallinity. The peaks of diffraction angles represent crystal phase and the crystallinity is characterized by the strength of peak. High and narrow peaks represent high crystallinity and stable crystal structure. We expected to obtain a new crystalline phase with compact physical structure and stable chemical properties via encapsulation. According to the results of XRD, we actually obtained the new crystalline phase for GLOM expected, which is a compact structure with KGM/SPI wall encapsulating GLO. The formation of new crystalline phases also proved GLO was successfully be encapsulated by KGM and SPI. We also have carefully revised our manuscript. See the line 337-340 and 346-348.
Q13: In the part 3.4, “Morphology of microcapsules”, before the part [lines 313-314] “the morphology of the KGM and SPI wall materials exhibited a sheet form with serious dents and folds, please clearly indicate how these KGM + SPI samples have been prepared by freeze-drying. Normally, by freeze-drying a homogeneous KGM + SPI solution one may expecting to obtain flakes.
Response13: Thanks very much for your professional advice. The preparation of KGM/SPI wall has been added in the material and method. See the line 107-111 and 123-127.
2.2.2.1 Preparation of KMG/SPI solution
KGM/SPI solution was prepared from KGM (7.0 g) and SPI (2.4 g) in distilled water and constantly stirring for 10 min at 50℃, which has been obtained according to trial planning. Then the emulsion was homogenized by a high-speed dispersator (XHF-D, Xinzhi Corp, Ningbo, China) at 10000 rpm for 5 min.
2.2.3 Freeze drying
The KGM/SPI solution and KGM/SPI/GLO emulsion were precooled at -80℃ and then put into the freeze-dryer to dry for 36 h to obtain dried KGM/SPI wall and GLOM. The prepared samples were ground into powder and stored in a plastic container at 4℃ for further experiments.
Q14: Please reformulate and explain the lines 333-337: “Besides, previous studies reported that core shell structure helped to increase the encapsulation efficiency of microcapsules. Besides, wall materials with even thickness properties helped to enhance its oxidative stability and to increase the mechanical strength of microcapsules [23], which indicated that GLOM might have excellent oxidative stability.
Response14: Thanks very much for your advice. The sentence has been revised into “Besides, previous studies reported that the structure of shell encapsulating core helped to increase the encapsulation efficiency of microcapsules. Besides, wall materials with homogeneous thickness could form a physical barrier, which might contribute to enhance its oxidative stability and to increase the mechanical strength of microcapsules” in our manuscript in the line 381-385.
Q15: Please explain the meaning of the comparison from lines 352-354, of the bulk density of GLOM with the ones of pumpkin seed oil microcapsules and flaxseed oil microcapsules (all obtained by freeze drying), since the bulk density of one kind of core-shell microcapsules also depends on the bulk densities of the components, the core/shell ratio, shell thickness.
Response15: Thanks very much for your professional question. See the line 403-407.
Bulk density is a momentous parameter during the processing, storage and transportation conditions of powdered food. The bulk density of GLOM was larger than pumpkin seed oil microcapsules and flaxseed oil microcapsules, indicating KGM/SPI wall has advantages during the processing, storage and transportation conditions. Samples with the lower bulk density values are more sensitive to oxidative degradation during storage because there are more air spaces between particles [40]. Therefore, the components we selected might contribute to oxidative stability of GLOM We have carefully revised our manuscript.
Q16: Please explain which is the practical interest of the DSC study on these microcapsules vs. individual components and the interest of the different phase transitions identified by DSC.
Response16: Thanks very much for your professional question. Differential scanning calorimetry can be used to detect the transformation heat of sample in a range of temperature, which can determine the thermal stability of sample.
DSC study has proved that GLOM remain stable below 62℃. Generally, the temperature of manufacture process is below 62℃. Therefore, the physicochemical property of GLOM will not change and the nutrition of GLOM will not degrade during the manufacture process. In addition, the phase transition of free GLO happened from 25℃ but the phase transition of GLOM happened from 62℃. DSC study on individual components and different phase transitions proved that compared with GLO, the thermal stability of GLOM improved after encapsulation by KGM/SPI wall. We have revised our manuscript carefully. See the line 436-437 and 448-452.

Round 2
Reviewer 2 Report
The revised version of the manuscript foods-1678321 is highly improved. The Authors have answered and corrected most of my concerns.
I found that the Authors correctly answered to most of my concerns.
The changes they made to the text proved to me that they have a good knowledge of the subject and that the problems were mainly related to how to write and communicate their research.
Hoping my review could help